# Shade suppresses wound-induced leaf repositioning through a mechanism involving *PHYTOCHROME KINASE SUBSTRATE* (*PKS*) genes

**Anne-Sophie Fiorucci**[1¤a], **Olivier Michaud**[1], **Emanuel Schmid-Siegert**[2¤b], **Martine Trevisan**[1], **Laure Allenbach Petrolati**[1], **Yetkin Çaka Ince**[1¤c], **Christian Fankhauser**[1]*

**1** Faculty of Biology and Medicine, Centre for Integrative Genomics, University of Lausanne, Lausanne, Switzerland, **2** SIB-Swiss Institute of Bioinformatics, University of Lausanne, Lausanne, Switzerland

¤a Current address: Université Paris-Saclay, CEA, CNRS, Institute for Integrative Biology of the Cell (I2BC), Gif-sur-Yvette, France
¤b Current address: JRS-NGSAI, Epalinges, Switzerland.
¤c Current address: Department of Plant and Microbial Biology, University of Zürich, Zürich, Switzerland.
* Christian.fankhauser@unil.ch

**Data Availability Statement:** RNA-seq data reported in this article have been deposited in NCBI's Gene Expression Omnibus and are

## Abstract

Shaded plants challenged with herbivores or pathogens prioritize growth over defense. However, most experiments have focused on the effect of shading light cues on defense responses. To investigate the potential interaction between shade-avoidance and wounding-induced Jasmonate (JA)-mediated signaling on leaf growth and movement, we used repetitive mechanical wounding of leaf blades to mimic herbivore attacks. Phenotyping experiments with combined treatments on *Arabidopsis thaliana* rosettes revealed that shade strongly inhibits the wound effect on leaf elevation. By contrast, petiole length is reduced by wounding both in the sun and in the shade. Thus, the relationship between the shade and wounding/JA pathways varies depending on the physiological response, implying that leaf growth and movement can be uncoupled. Using RNA-sequencing, we identified genes with expression patterns matching the hyponastic response (opposite regulation by both stimuli, interaction between treatments with shade dominating the wound signal). Among them were genes from the *PKS* (Phytochrome Kinase Substrate) family, which was previously studied for its role in phototropism and leaf positioning. Interestingly, we observed reduced shade suppression of the wounding effect in *pks2pks4* double mutants while a *PKS4* overexpressing line showed constitutively elevated leaves and was less sensitive to wounding. Our results indicate a trait-specific interrelationship between shade and wounding cues on Arabidopsis leaf growth and positioning. Moreover, we identify *PKS* genes as integrators of external cues in the control of leaf hyponasty further emphasizing the role of these genes in aerial organ positioning.

## Author summary

Plants face different types of stressful situations without the ability to relocate to favorable environments. For example, increasing plant density reduces access to sunlight as plants

accessible through GEO Series accession number GSE133252.

**Funding:** This project was funded by the University of Lausanne and the Swiss National Science Foundation (grants no. CRSII3_154438, 310030_200318 and 310030B_179558 to C.F.). The funders had no role in study design, data collection and analysis, decision to publish, or preparation of the manuscript.

**Competing interests:** The authors have declared that no competing interests exist.

start to shade each other. Foliar shading represents a stress that many plants cope with by changing their morphology. This includes elongation of stem-like structures and repositioning of leaves to favor access to unfiltered sunlight. Plants also defend themselves against various pathogens including herbivores. Defense mechanisms include the production of deterrent chemical and morphological adaptations such as stunted growth and downwards leaf repositioning. Here we studied the morphological response of plants when simultaneously facing shade and herbivore stress. When facing both stresses petiole growth was intermediate between the shade-enhanced and wound-repressed response. In contrast, the shade cue overrides the wounding cue leading to a similar upwards leaf repositioning in the combined treatments or in the response to shade alone. Using gene expression analyses and genetics we identified two members of the Phytochrome Kinase Substrate family as playing a signal integration role when plants simultaneously faced both stresses. This contributes to our understanding of the mechanisms underlying plant morphological adaptations when facing multiple stresses.

## Introduction

Plants constantly adjust their growth and development in response to variations in abiotic and biotic environmental parameters. Under competition for light resources encountered in dense communities, shade-avoiding plants initiate a series of developmental changes known as the shade-avoidance syndrome (SAS). At the vegetative stage, this response is characterized by elongation of aerial organs like hypocotyls, petioles and stems and upward leaf positioning, also known as hyponasty [1]. SAS-associated phenotypes help plants overtop competitors, get better access to sunlight and are ultimately associated with better fitness [1,2]. Detection of competitive neighbors occurs through a change in light quality reaching the plant. Indeed, because of the spectral properties of leaves, plants reflect far-red (FR) wavebands, which leads to an increased amount of FR and a decrease of the red (R) to FR ratio (low R/FR) [2]. Changes in the R/FR ratio are perceived by phytochrome photoreceptors, especially phyB, and serve as a primary signal of vegetation proximity. Under low R/FR conditions, phyB is converted from the active Pfr to the inactive Pr form. Phytochrome photoconversion to the Pr form allows the stabilization and/or activation of transcription factors from the PIF family (Phytochrome-Interacting Factors) [3]. PIFs, especially PIF4, PIF5 and PIF7, play a major role in the reprogramming of gene expression during shade-avoidance, notably by inducing genes controlling auxin homeostasis [4]. At the rosette stage, these three PIFs control shade-induced petiole elongation [5] and hyponasty [6,7].

Herbivore attack induces an array of defense mechanisms depending on the Jasmonate (JA) pathway [8]. JA is a lipid-derived hormone synthesized from α-linolenic acid. Increased JA levels are perceived by the SCF$^{COI1}$-JAZ co-receptor. This leads to the degradation of JAZs repressors by the 26S proteasome, which releases the activity of downstream transcription factors from the MYC family, especially MYC2, as well as MYC3 and MYC4 [9]. MYCs are responsible for a large part of JA-dependent transcriptional reprogramming. Interestingly, JA-induced defenses are activated both in the eaten organ and in distal tissues. In rosettes, information about an ongoing attack is transmitted to vascularly-connected leaves through propagation of electrical signals [10,11], warning and preparing intact tissues for a potential aggression. Mechanical leaf wounding, using toothed forceps for example, mimics the effect of chewing by herbivores and is sufficient to induce JA production and the induction of defense mechanisms, both in harmed and distal tissues [12]. Induction of JA-dependent defense

comes with a strong inhibition of growth [13], at least partly due to an inhibition of cell prolif-
eration in leaves [14,15]. The effect of JA or wounding on leaf position is less understood, but
several studies suggest that wound-response signaling inhibits circadian leaf movements
[16,17].

Plants often face multiple challenges and must integrate this information to respond
accordingly. For example, in the growth-defense tradeoff shaded plants challenged with patho-
gens prioritize growth over defense, making them more susceptible to biotic stresses [4,18].
This tradeoff was long thought to be based on resource allocation but it can also be explained
by tightly controlled antagonistic transcriptional networks [19,20]. Indeed, it is possible to
genetically uncouple growth and defense and obtain plants that are able to both grow and
defend to high levels [21,22]. The uncoupling of growth and defense depends on the level of
defense [23]. In the case of shade conditions, various mechanisms contribute to the attenua-
tion of JA-dependent defense. Low R/FR ratios favor the stability of JAZ repressors through
degradation of DELLA proteins [24], and also decrease the stability of MYC transcription fac-
tors [25]. In parallel, shade signals directly decrease the production of active JA through the
PIF-dependent induction of *ST2a*, which codes for a sulfotransferase responsible for reducing
the pool of precursors of active JAs [26].

While the impact of shading conditions on JA-dependent defense responses has been
extensively studied, our current knowledge of how plants integrate these two pathways at the
level of growth remains limited. A previous study showed that exogenous methyl-jasmonate
(MeJA) application does not prevent low R/FR-induced petiole elongation [27]. In contrast,
shade-mimicking conditions suppressed MeJA-dependent inhibition of hypocotyl growth in
young Arabidopsis seedlings [24]. These observations indicate a complex relationship between
shade and the JA pathway during growth regulation. Moreover, how shade and jasmonate reg-
ulate leaf movements remains unknown. Understanding the potential interaction between
shade-avoidance and the JA pathway on leaf growth and hyponasty can provide new informa-
tion about these fundamental growth processes. Here we aimed at understanding how plants
integrate environmental signals with opposing effects on leaf growth and movement, using
shade and wounding as opposite cues.

## Results

### Shade suppresses the wounding effect on leaf elevation but not on elongation

We first defined precise protocols for shade and wounding treatments. For shade treatment,
we used previously described conditions [5], with shade (or control light) treatment applied
for three days to 17-day-old, long-day grown plants. Our shade conditions were a reduction in
the red to far-red ratio (low R/FR), while maintaining PAR constant [5]. In these conditions,
leaves 3 and 4 show pronounced petiole elongation upon exposure to shade, whereas leaves 1
and 2 barely respond to shade [5]. In parallel, we chose repetitive mechanical wounding to
induce JA production in a systemic way. We used toothed forceps to crush the apical half of a
leaf blade [11]. In 4-week-old, short-day-grown plants, one such wound on leaf 8 is sufficient
to induce the JA pathway in leaves connected by the vasculature, that is leaves 5, 11, 13 and 16
[11]. In our growth conditions, we observed that wounding of leaf 1 rapidly induced expres-
sion of the JAZ10p:GUSPlus (JGP) reporter [28] in all leaves, especially leaves 3, 4 and 5 (S1A
Fig), as well as other JA-marker genes in leaf 4 (S1B Fig). We thus concluded that at this devel-
opmental stage, we could induce the JA pathway in all intact leaves by wounding leaf 1. Previ-
ous studies have shown that serial wounding of different rosette leaves was efficient to trigger a
growth defect of all leaves [13], especially on petioles [29]. We therefore used an adapted serial

wounding protocol, with three successive wounds on leaf 1, 2, 3 at one-day intervals and measured leaf position and petiole growth of leaf 4 (Fig 1A).

To determine whether wounding influences shade-induced leaf growth and movement, we pre-treated plants with wounding and then started shade treatment right after the 3rd wound (Fig 1A). We compared the effect of wounding, low R/FR and the combined treatments. We measured the elevation angle of the first intact leaf (leaf 4) at the end of day 18, and petiole length of the same leaf after 3 days (Fig 1A). As shown by histochemical detection of cyclin activity, there was barely any cell division occurring in leaf 4 at the start of the wounding treatment (S1C Fig). Therefore, leaf 4 petiole growth mainly depended on cell elongation in these experimental conditions. As expected, the shade treatment induced both leaf elevation and petiole elongation in wild-type plants (Fig 1B). In contrast, wounded plants showed a significantly lower leaf position and shorter petioles, which is consistent with previous reports [13,15,17,22,29]. The effect of wounding was largely diminished in an *aos* (Allene Oxide Synthase) mutant background (Fig 1B), which is deficient in JA production [30]. The wounding effect on leaf growth and movement observed here was thus jasmonate-dependent.

Upon combined treatments, we observed that the negative effect of wounding on leaf elevation angle was largely suppressed in shade-treated wild-type plants (Fig 1B and 1C). There was an interaction between shade and wounding on the regulation of leaf hyponasty (Fig 1B and 1C), similar to the inhibition of defense responses by shade-avoidance (Light x Wound interaction term in a two-way ANOVA on wt plants was significant, with p-value $< 2.10^{-16}$). In contrast, we did not observe any interaction on petiole elongation (Light x Wound interaction term in a two-way ANOVA on wt plants was not significant). This result is consistent with a previous study on petiole elongation combining shade and exogenous MeJA application [27]. We changed the timing of wounding versus shade (either started at the same time or one day after the start of shade treatment) and observed the same trend (S2 Fig). We concluded that there is no interaction between shade and wounding on petiole elongation, which contrasts with the inhibition of defense responses by shade-avoidance. Consistent with our data in the wild type, in a *phyB* mutant, which displays a constitutive shade-avoidance phenotype under sun-mimicking conditions, wounding diminished petiole growth but not leaf hyponasty (S3 Fig).

Shade-induced hyponasty depends on *YUCCA* genes [6], which code for the rate-limiting enzymes of auxin biosynthesis. In particular, *YUC2*, *YUC5*, *YUC8* and *YUC9* are up-regulated in low R/FR conditions [31] and the corresponding quadruple *yuc2589* mutant is unresponsive to shade [6,31]. We observed no suppression of the wounding effect on hyponasty by shade in the *yuc2589* background (Fig 1C). This indicates that YUC activity upon shade perception is necessary to suppress the wounding effect on leaf hyponasty. Collectively, these data indicate that low R/FR (shade) suppressed the effect of wounding on leaf hyponasty, while both treatments affected growth in an additive way. Moreover, the effect of wounding on hyponasty depended on JA signaling, while low R/FR-induced auxin production was required to suppress the wounding effect.

Because the measure of leaf elevation was done at a defined time (ZT11), we could not rule out the possibility that wounding delays the circadian leaf movement instead of suppressing it. To study the kinetics of the hyponastic response we took advantage of a phenotyping system allowing the determination of leaf elevation with high temporal resolution [6,32]. By applying the same protocol as before (Fig 1A), we first observed that wounding inhibits the diurnal circadian movement of distal leaf 4 under normal light conditions (Fig 1D, left panel). However, leaf 4 recovered a normal movement one day after the last wound, suggesting that the impact of wounding on leaf elevation is transient. As observed previously, shade rapidly triggered an upwards repositioning of leaf 4 (Fig 1D) [6]. Moreover, in low R/FR, wounding hardly affected

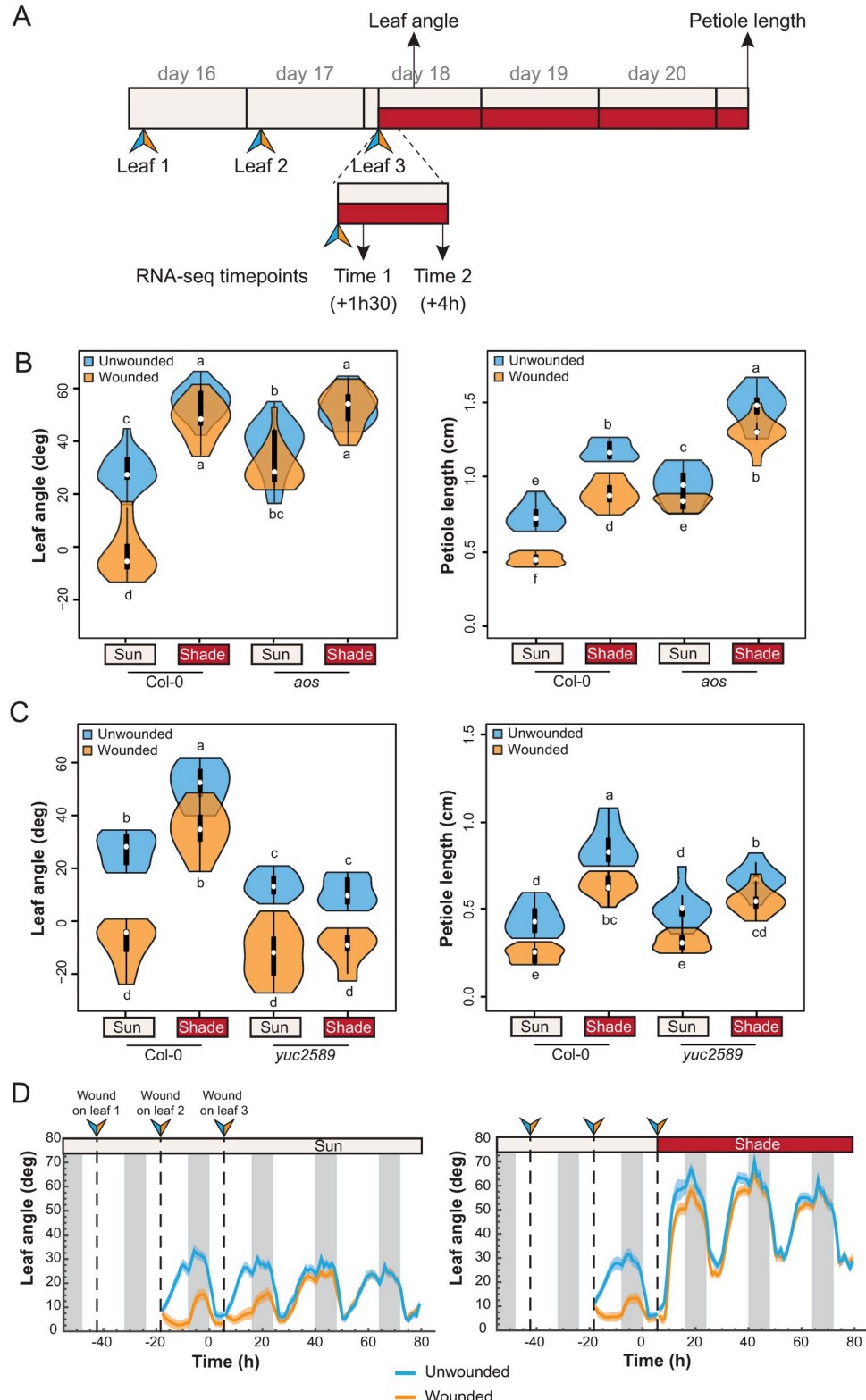

**Fig 1. Shade suppresses the wounding effect on leaf elevation but not on elongation. A)** Schematic of the experimental procedure. Wounds are performed individually on leaf 1, then leaf 2 and the shade treatment starts following wounding of leaf 3. We measured elevation angle of leaf 4 at the end of day 18 on pictures of each individual plant and petiole length of the same leaf at the end of the treatments (day 21). See details in Material and Methods. **B)**

Leaf angle and petiole length phenotype of wt Col-0 and *aos* mutant plants (n = 10 plants/genotype/condition). Representative experiment from 3 biological replicates. **C)** Leaf angle and petiole length phenotype of wt Col-0 and *yuc2589* mutant plants (n = 10 plants/genotype/condition). Representative experiment from 2 biological replicates. **D)** Elevation angle of leaf 4 in unwounded and wounded wt Col-0 plants either kept in normal light conditions (left panel) or shifted to shade after the third wound (right panel). Grey vertical bars represent night periods. Each graph corresponds to a representative experiment from 2 biological replicates (n>35 plants/genotype/condition). In B) and C), graphs are represented as violin plots, which present a combination of a box plot and a kernel density plot. In each box plot, the white dot represents the median, black boxes extend from the 25th to the 75th percentile, while the vertical black line extends to 1.5 times the interquartile range of the lower and upper quartiles, respectively. A rotated kernel density plot surrounds each side of the box plot. Different letters indicate significant differences (Tukey's HSD test following a three-way ANOVA, P < 0.05).

the hyponastic response (Fig 1D, right panel). Taken together our phenotyping experiments revealed that leaf growth and leaf movement can be at least partially uncoupled in the context of combined shade and wounding treatments.

## Shade and wounding lead to contrasted gene expression reprogramming

We analyzed the gene expression pattern of petioles from leaf 4 in single and combined treatments, with the experimental design based on our phenotyping tests (Fig 1A). Samples for genome-wide gene expression analysis using RNA-sequencing were collected on day 18, 1.5h and 4h after the start of the treatments (wound/mock & high/low R/FR). Both time points were chosen according to the kinetics of movement as determined by the phenotyping experiments (Fig 1D). To distinguish jasmonate-dependent effects, the experiment was conducted on wild-type Col-0 and *aos* mutant plants.

We first looked for differentially expressed genes (DEGs) upon shade treatment by comparing low R/FR-treated to high R/FR-treated samples in the different conditions (Fig 2). We observed that most of the variation in gene expression is visible at the second time-point (Time 2, 4h, Fig 1A), as seen by the number of DEGs (Fig 2A) and the stronger fold change observed in the hierarchical clustering (Fig 2B). This result is consistent with previous studies done in young seedlings [31]. Overall, there was no major difference in the total number of shade-regulated genes between wounded or unwounded plants (Fig 2A). However, at each time-point the wounded Col-0 sample showed a divergent pattern compared to the three others (Fig 2B). These experiments showed that the wounding treatment altered shade-induced gene expression reprogramming in a JA-dependent manner.

We then analyzed the effect of wounding on gene expression (S4 Fig). As expected, in the *aos* mutant there was almost no response to wounding at the gene expression level. Therefore, most of the wound-dependent gene regulation in this context was JA-dependent. The wound response appeared to show faster kinetics compared to the shade response, with a high number of DEGs already at Time 1 (S4A Fig). However, the first wound was performed two days before the shade treatment (Fig 1A), which makes it difficult to compare the kinetics of both responses. Interestingly, the shade treatment had a major impact on wound-regulated gene expression with: (1) a lower number of DEGs in shade compared to sun conditions (S4A Fig) and (2) an altered pattern of expression in shade vs sun at both time-points (S4B Fig). By looking at all the wound response DEGs at Time 1 and/or Time 2, we confirmed that 41% of wound-DEG in sun were not significantly differentially regulated by wounding in shade conditions (890 genes, S4C Fig). Besides, for those that were still regulated in shade (1278 genes), they tended to be regulated to a lesser extent compared to sun conditions (S4D Fig). Therefore, under shade-mimicking conditions, there was a global attenuation effect of the wound-regulated gene expression program, which is consistent with the inhibition of defense by shade as well as the suppression of the wounding effect on leaf elevation.

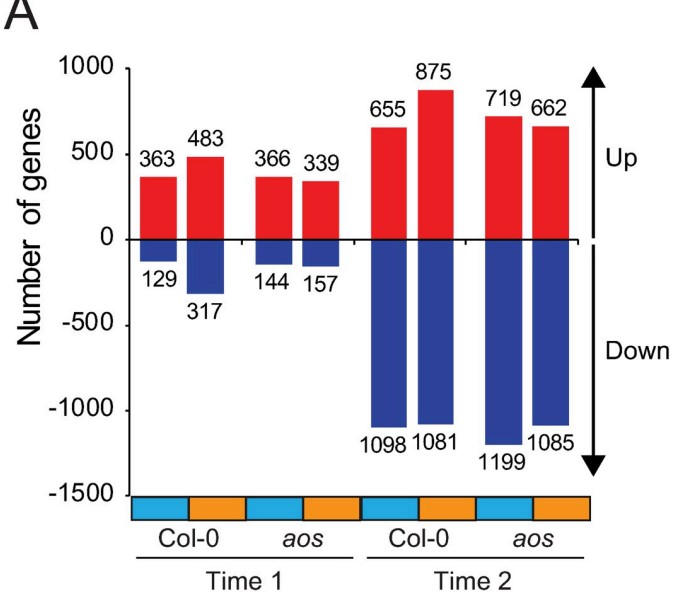

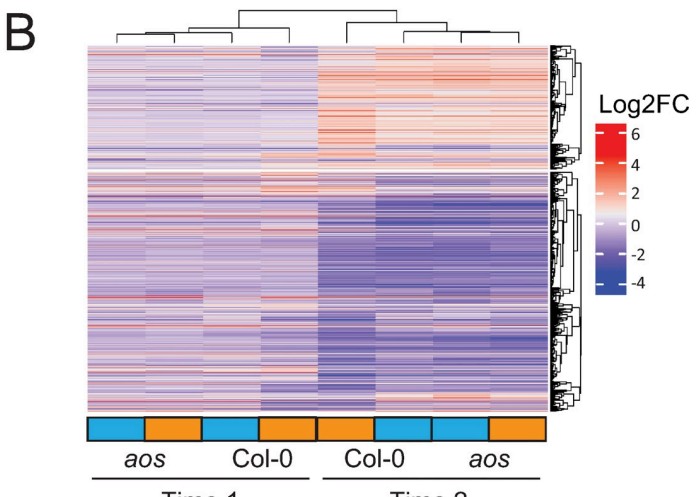

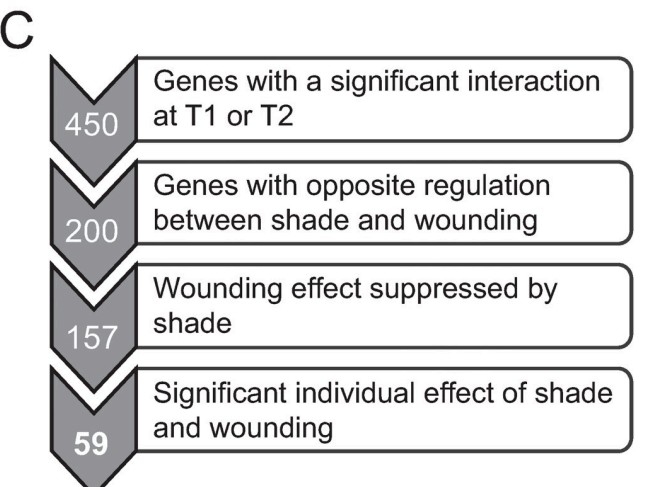

**Fig 2. Shade-regulated gene expression in petioles. A)** Number of genes significantly differentially regulated by shade in the different conditions: genotypes (Col-0 and *aos*), timepoints (Time 1 and Time 2) and wounding treatment (unwounded in blue, wounded in orange). Genes are considered differentially expressed when |log2 fold change|>1 and pval < 0.05. **B)** Hierarchical clustering of significantly differentially regulated genes by shade in the different conditions (same genes as in A). **C)** Schematic representation of the successive criteria applied to find candidate genes involved in the shade-wounding interaction on hyponasty. Numbers of genes are indicated on the left.

## *PKS* genes regulate leaf hyponasty during combined shade-wounding treatments

We hypothesized that genes with an expression pattern similar to the hyponastic response pattern (opposite regulation by both stimuli, interaction between treatments with shade dominating the wound signal) may be important for the regulation of leaf movement and possibly the interaction between shade and wounding. We thus made a list of genes matching the following criteria: (1) significant interaction between shade and wounding at least at one time point (450 genes), (2) opposite regulation by the individual treatments (200 genes), (3) wounding effect suppressed by shade upon combined treatments (157 genes) and (4) significant effects of individual shade and wounding treatments (59 genes) (Fig 2C). This final list contains many genes involved in auxin, GA or brassinosteroids biosynthesis and signaling pathways (S1 Table).

One gene in the shortlist seemed particularly interesting to test as a candidate: *PKS4* (Phytochrome Kinase Substrate 4, At5g04190). Indeed, *PKS4* is induced by shade, repressed by wounding and its expression level was similar to shade alone upon combined treatments (S5A Fig). *PKS4* belongs to a family of four genes which have been mostly studied for their role in phototropin signaling, and *PKS1*, the founding member of this family, was first shown to interact with phytochromes [33–39]. Using a PKS4::GUS reporter line, we confirmed the induction of *PKS4* expression by shade in 3-week-old rosettes (S5B Fig). Interestingly, *PKS4* was mostly expressed in the petioles of younger leaves as well as in the mid-vein. However, a *pks4* loss-of-function mutant was not significantly different from the wild type during shade, wounding or combined treatments both for leaf angle and petiole length (S5C Fig).

Among the three other genes of the family, *PKS2* showed a similar expression pattern but was not present in the list of candidates because it did not reach the threshold of significance for the interaction (S6A Fig). Interestingly, *PKS2* was shown to be important for leaf positioning [33]. Whereas a *pks2* single mutant showed wild-type phenotypes (S6B Fig), we observed reduced shade suppression of wounding-triggered leaf elevation in a *pks2pks4* double mutant (Fig 3A; Tukey HSD group b for Col-0 upon combined treatment, group c for *pks2pks4*). In contrast, responses to individual shade or wounding treatments were similar to the wild type (similar Tukey HSD groups for individual treatments), as well as petiole length phenotypes in all tested conditions (Fig 3A). *PKS2* and *PKS4* therefore act to control hyponasty during combined shade and wounding treatments. It is noteworthy that petiole length responses were not affected in this mutant background, suggesting that the effect of *PKS* genes is indeed specific to leaf movement. We confirmed these results in another *pks2 pks4* allelic combination (S7 Fig). We then tested a quadruple *pks1234* mutant [40]. In this background, there was even less suppression by shade of the wounding effect on leaf position than in the double mutant, highlighting redundancy among *PKS* genes (Fig 3B). However, this mutant also showed reduced petiole elongation in all tested conditions (Fig 3B). In addition to comparing the response of all genotypes in different conditions, we determined how the interaction between treatments was affected in the different genotypes by computing the three-way ANOVA interaction term (Genotype x Light x Wounding = GxLxW) (S2 Table). In *pks1234*, the p-value associated with the GxLxW for tip angle was largely below 0.05 (S2 Table), indicating that in this mutant the interaction between treatments (wound and shade) was different from the wild type. This was

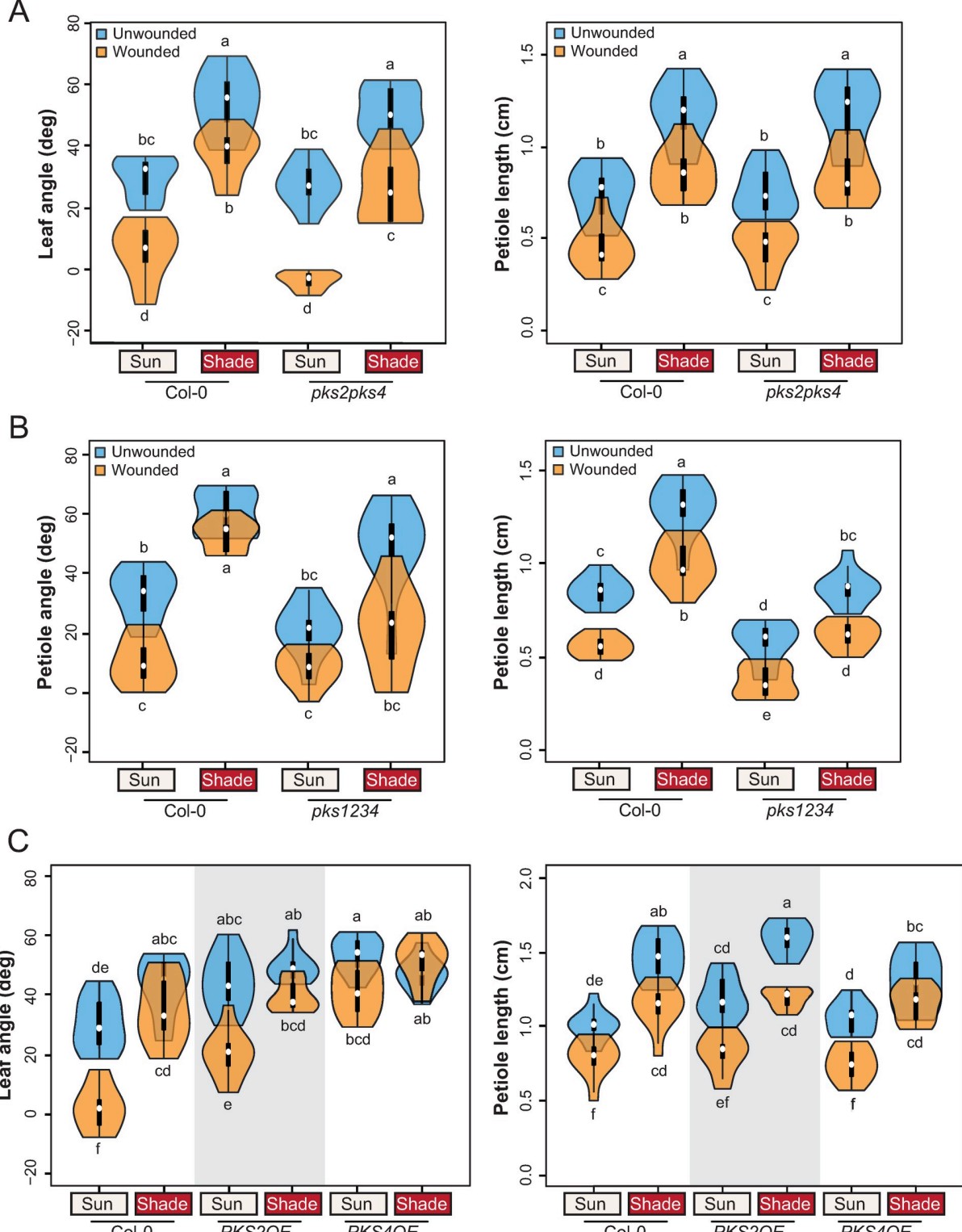

**Fig 3. *PKS* genes regulate the shade-wounding interaction on hyponasty. A)** Leaf angle and petiole length phenotype of wt Col-0 and *pks2-1pks4-1* mutant plants (n = 10 plants/genotype/condition). Representative experiment from 3 biological replicates. **B)** Petiole angle and petiole length phenotype of wt Col-0 and *pks1-1pks2-1pks3-9pks4-1 (pks1234)* mutant plants (n = 10 plants/genotype/condition). The quadruple *pks1234* mutant displays a leaf flattening defect and thus only petiole angles could be measured for this genotype. Representative experiment from 2 biological replicates. **C)** Leaf angle and petiole length phenotype of wt Col-0, *PKS2OE* and *PKS4OE* plants (n = 10 plants/genotype/

condition). Representative experiment from 2 biological replicates. All experiments were done as described on Fig 1A. Different letters indicate significant differences (three-way ANOVA with Tukey's HSD test, $P < 0.05$). All graphs are represented as violin plots, which present a combination of a box plot and a kernel density plot. In each box plot, the white dot represents the median, black boxes extend from the 25th to the 75th percentile, while the vertical black line extends to 1.5 times the interquartile range of the lower and upper quartiles, respectively. A rotated kernel density plot surrounds each side of the box plot.

not observed in *pks2pks4* double mutants (both alleles), possibly because of (1) the limited number of plants used in each experiment and (2) the smaller effect observed in the double *pks2pks4* mutant compared to *pks1234*. We therefore conclude that in the *pks1234* quadruple mutant both the hyponastic response in combined treatment and the interaction between treatments was different from the wild type, while in *pks2pks4* only the response to combined treatments was significantly different from the wild type. Finally, a *PKS4* overexpressing line showed constitutively elevated leaves and was less sensitive to wounding (Fig 3C). Taken together these results confirm that *PKS4* and *PKS2* are required for shade-induced suppression of the wound-triggered inhibition of leaf hyponasty. In contrast, in the conditions tested, these genes were not required for the response to separate treatments.

## *PKS* genes are induced by shade in a PIF-dependent manner

The importance of *PKS2* and *PKS4* and their normal expression in the control of leaf hyponasty (Fig 3) prompted us to analyze the mechanisms underlying their expression. Because PIFs are (1) the main transcription factors involved in the regulation of gene expression in response to shade and (2) required for a proper shade-induced hyponasty [6,7], we tested the induction of *PKS* genes by shade in a *pif457* triple mutant [5]. In response to 1h30 shade treatment, *PKS4* and *PKS2* were both significantly induced in leaf 4 of 3-week-old wild-type rosettes, but not in the *pif457* background (Fig 4A). Expression of *YUC8* was used as a positive control for this experiment. Thus PIF4, 5 and/or 7 are required for the induction of *PKS4* and *PKS2* by shade. By browsing published ChIP-seq datasets, we found that the *PKS4* promoter is bound by PIF4 in various experimental conditions [41,42] in a region containing a PBE-box (Fig 4B). To test whether PIF4 directly regulates *PKS4* expression under low R/FR conditions, we performed a ChIP-qPCR experiment using a PIF4-HA tagged line [43]. We observed that PIF4-HA was bound to the *PKS4* promoter region after 2h of shade treatment in 10-day-old seedlings, although to a lower extent than to *PIL1*, a known PIF-target (Fig 4B). This might be due to the restricted expression of *PKS4* in shade-treated seedlings (S5B Fig). We therefore conclude that PIF4 directly regulates the induction of *PKS4* upon shade treatment. To determine whether the PIFs are required for the shade-induced suppression of the wounding effect on leaf position, we analyzed petiole length and hyponasty in the *pif457* triple mutant. Consistent with previous publications *pif457* mutant plants did not elevate their leaves in response to shade treatments (Fig 4C) [6,7]. However, the *pif457* mutant responded similarly to the wild type to the wound treatment (Fig 4C). Moreover, in *pif457* the inhibition of leaf elevation by wounding was still visible upon combined treatment. This shows that PIFs and YUCs are required for the shade-induced suppression of the wounding effect on hyponasty (Figs 1C and 4C).

## *PKS4* and *PKS2* are repressed by MYCs in response to wounding

To understand how *PKS* genes were regulated by wounding, we looked at their expression in the *aos* mutant from our RNAseq dataset. We observed that the repression of both *PKS4* and *PKS2* by wounding under sun-mimicking conditions was not visible in the *aos* mutant (S8 Fig). This result is consistent with previous reports showing a down-regulation of *PKS2* and

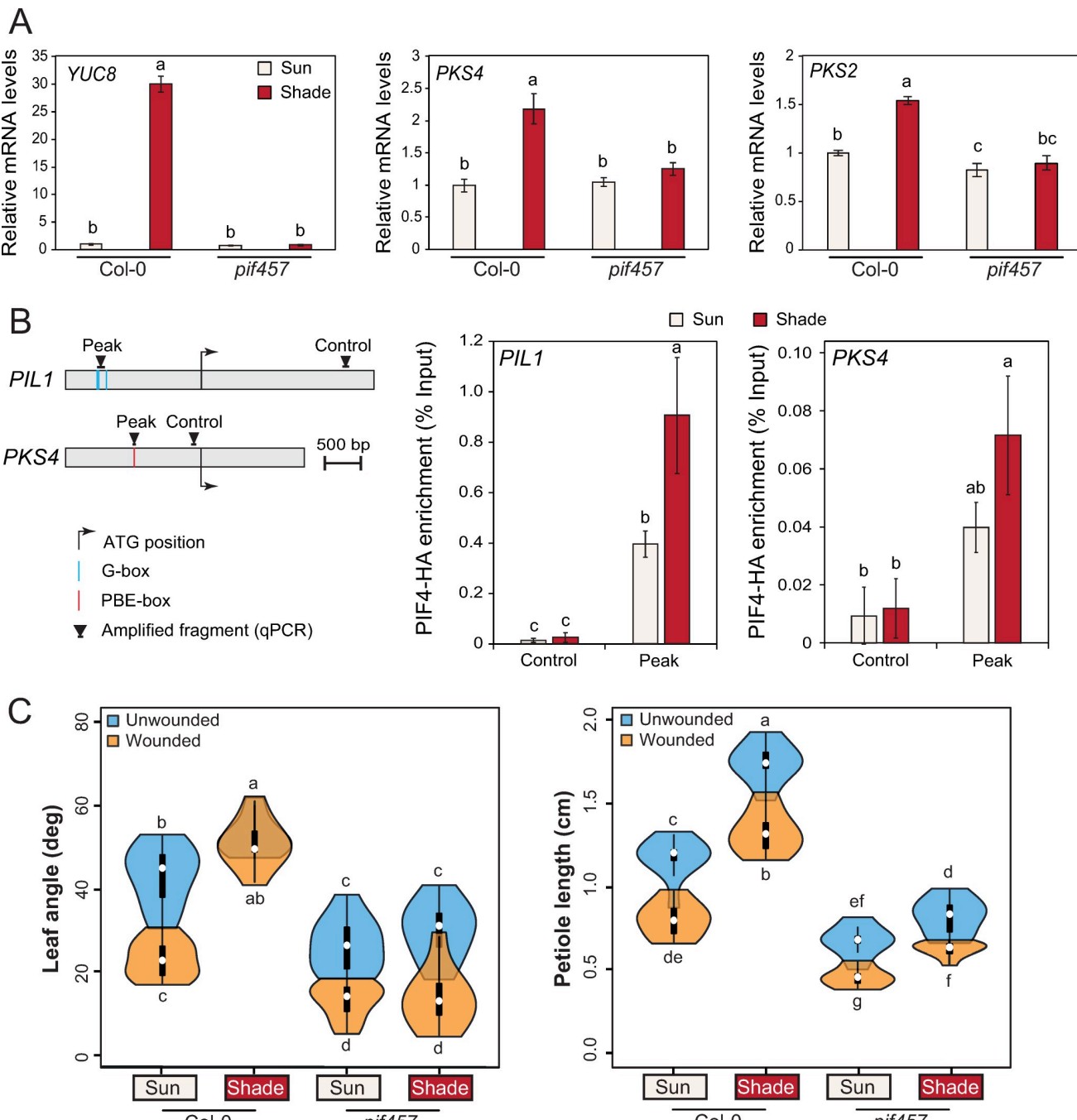

**Fig 4. *PKS4* and *PKS2* are induced by shade in a PIF-dependent manner. A)** Relative expression of *YUC8*, *PKS4* and *PKS2* evaluated in entire leaf 4 from 18-d-old wt Col-0 and *pif457* mutants either kept in high R/FR (sun) or transferred to low R/FR (shade) at ZT3 for 90 minutes; samples in sun and shade were harvested at the same ZT. Gene expression values were calculated as fold induction relative to a Col-0 sample in high R/FR. n = 3 (biological) with three technical replicas for each RNA sample. Data are means ± 2 SE. Different letters indicate significant differences (P < 0.05). **B)** PIF4-HA binding to the promoter of *PIL1* and *PKS4* evaluated by ChIP-qPCR in 10-d-old seedlings either kept in high R/FR or transferred for 2 h to low R/FR at ZT2. Input and immunoprecipitated DNA were quantified by qPCR using primers shown on the schematic representation of the genes with 'Peak' indicating where PIF4 binding was identified before (top panel). PIF4-HA enrichment is presented as IP/Input and error bars show standard deviation from three technical replicas. Different letters indicate significant differences (P < 0.05). Data from one representative experiment out of two biological replicates are shown. **C)** Leaf angle and petiole length phenotype of wt Col-0 and *pif457* mutant plants (n = 10 plants/genotype/condition). Representative experiment from 2 biological replicates. All experiments were done as described on Fig 1A. Different letters indicate significant differences (three-way ANOVA with Tukey's HSD test, P < 0.05). Graphs are represented as violin plots, which present a combination of a box plot and a kernel density plot. In each box plot, the white dot represents the median, black boxes extend from the 25th to the 75th percentile, while the vertical black line extends to 1.5 times the interquartile range of the lower and upper quartiles, respectively. A rotated kernel density plot surrounds each side of the box plot.

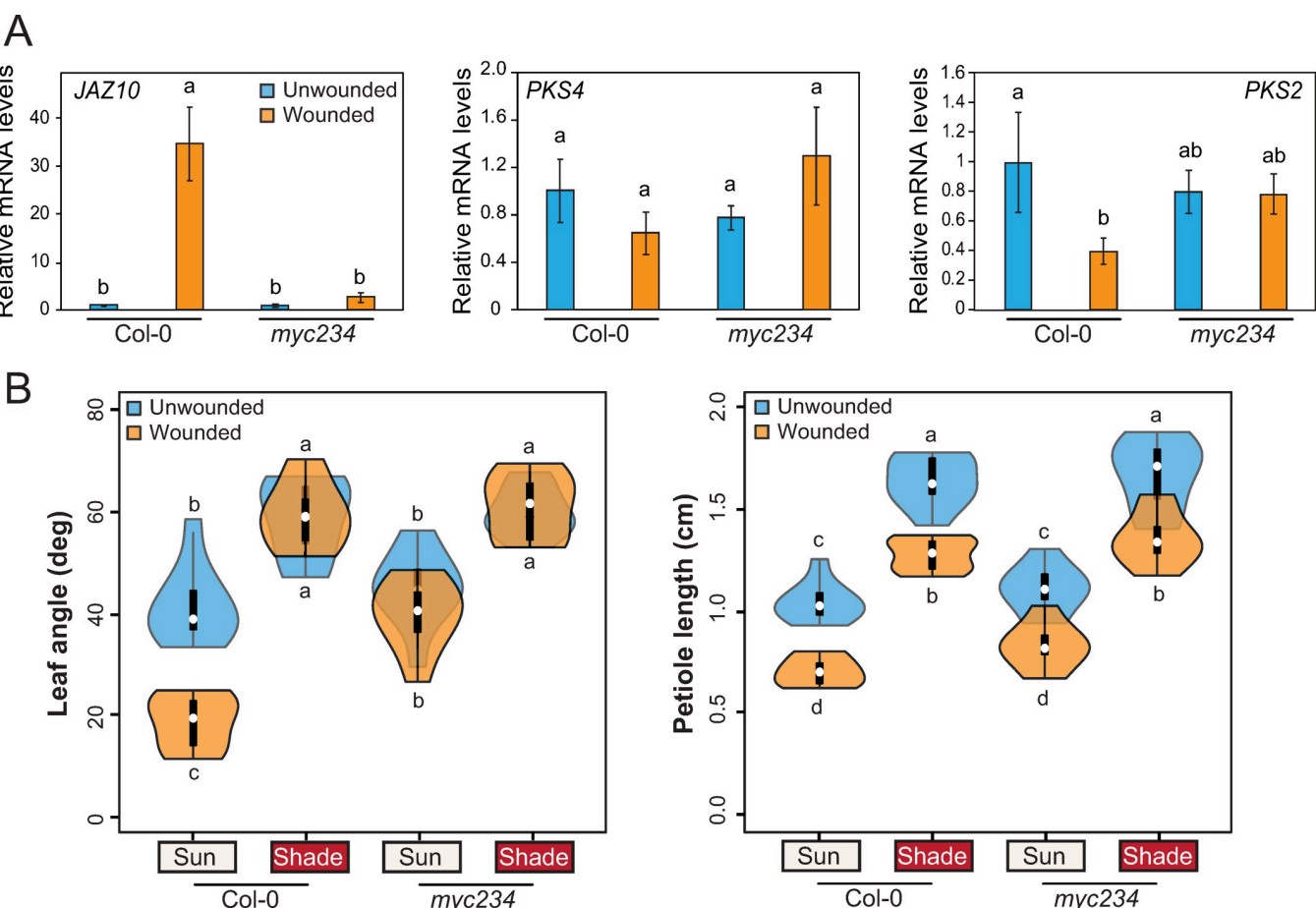

**Fig 5. *PKS4* and *PKS2* are repressed by MYCs in response to wounding. A)** Relative expression of *JAZ10*, *PKS4* and *PKS2* in leaf 4 petiole from 18-d-old wt Col-0 and *myc234* mutants either unwounded or wounded 3 times at ZT3 for 90 minutes; samples in sun and shade were harvested at the same ZT. Gene expression values were calculated as fold induction relative to a Col-0 sample in high R/FR. n = 3 (biological) with three technical replicas for each RNA sample. Data are means ± 2 SE. Different letters indicate significant differences (P < 0.05). **B)** Leaf angle and petiole length phenotype of wt Col-0 and *myc234* mutant plants (n = 10 plants/genotype/condition). Representative experiment from 2 biological replicates. All experiments were done as described for Fig 1A. Different letters indicate significant differences (three-way ANOVA with Tukey's HSD test, P < 0.05). Graphs are represented as violin plots, which present a combination of a box plot and a kernel density plot. In each box plot, the white dot represents the median, black boxes extend from the 25th to the 75th percentile, while the vertical black line extends to 1.5 times the interquartile range of the lower and upper quartiles, respectively. A rotated kernel density plot surrounds each side of the box plot.

*PKS4* upon wounding [44] or MeJA treatment [45,46] in green seedlings or leaves. We thus conclude that the impact of wounding on *PKS* expression depends on the jasmonate pathway. Since MYCs are the main transcription factors responsible for JA-dependent gene expression, we tested the expression of *PKS4* and *PKS2* upon wounding treatment in a *myc234* triple mutant [47]. As observed in our RNA-sequencing results, both genes were expressed at a lower level in the petiole of wounded wild types, although the difference was significant only for *PKS2* (Fig 5A). This may be explained by the modest repression of *PKS4* expression by wounding compared to *PKS4* induction by shade (S5A Fig). The repression of *PKS* gene expression by wounding was abolished in *myc234* petioles (Fig 5A), showing that MYCs are required for the repression of *PKS4* and *PKS2* upon wounding. To determine the phenotypic consequences of MYC-regulated transcriptional changes we characterized the *myc234* triple mutant (Fig 5B). The *myc234* mutant looked similar to the *aos* mutant in terms of leaf angle (Fig 1B), showing that the inhibition of leaf elevation by wounding depended on MYCs (Fig

5B). As reported previously [13], *aos* plants had longer petioles than the wt (Fig 1B), whereas *myc234* plants displayed similar petiole lengths to the wild type in all conditions (Fig 5B). We conclude that reduced *PKS2* and *PKS4* expression in response to wounding depends on JA production and MYC transcription factors. Moreover, JA and MYCs contribute to wound-regulated leaf hyponasty (Figs 1B and 5).

Based on our expression analyses, *PKS4* is thus regulated by both PIF4 and MYC2 (Figs 4 and 5). Because (1) PIFs and MYCs belong to the same family of transcription factors and (2) regulate similar phenotypic outputs, we hypothesized that they share target genes, and possibly regulate them in an opposite way, as for *PKS4*. We compared available ChIP-seq datasets for PIF4 [42] and MYC2 [48] which were obtained from plant material comparable to the one we used in this study. We observed a highly significant overlap between both gene lists (S9A Fig). Among our 59 candidate genes being regulated in opposite ways by shade and wounding with shade suppressing the wound effect (Fig 2C), 46 are up-regulated by shade and down-regulated by wounding, including *PKS4* (S1 Table). When comparing these 46 genes with ChIP-seq datasets, we observed that 69.6% (32 genes, including *PKS4*, significant overlap) are potentially direct targets of PIF4 and MYC2 (S9B Fig). Collectively this suggests that a large fraction of these genes are regulated in opposite ways by these bHLH transcription factors in response to shade and wounding with PIFs dominating the response in combined treatments.

## Discussion

In this study, we focused on the morphological consequences of combined shade and wounding treatments. In our experiments, shade and wounding did not show the same relationship for leaf hyponasty and petiole elongation. We observe that shade-mimicking conditions strongly inhibit downwards leaf repositioning induced by wounding (Fig 1). In contrast to leaf position, both signals act largely independently in the regulation of petiole length (Fig 1). This is consistent with earlier observations of wild-type petioles treated with low R/FR and MeJA [27] and with MeJA-treated *phyB* mutant plants [22]. However, our results contrast with the observation that a low R/FR treatment completely suppressed MeJA-induced inhibition of hypocotyl elongation [24]. Differences in developmental stages, experimental conditions or timing of the treatments might account for this discrepancy. In seedlings, MYCs directly regulate photomorphogenesis by inducing *HY5* expression [49], although further studies are needed to fully understand the interplay between MYCs and HY5 [50]. Recent phenotypic analyses of high-order *pif* and *myc* mutants coupled to transcriptomics also suggested that PIF and MYC transcription factors regulate petiole growth independently [51]. Under combined shade/wounding treatments, we interpret the difference between hyponasty and elongation as a partial uncoupling of growth and movement. Arabidopsis plants grown under different light regimes also show partially distinct growth and movement patterns [32]. Collectively, these findings imply that differential growth in petioles is not the only mechanism underlying leaf positioning.

The interaction between wounding and hyponasty is similar to defense responses with the shade response strongly attenuating the wound response (Fig 1). We observed that wounding transiently inhibits leaf elevation systemically in a JA-dependent manner. Moreover, the wound response was also strongly impaired in the *myc234* mutant (Figs 1 and 5). This is consistent with earlier reports showing that MeJA treatments and insect feeding trigger downward leaf movements [16,17]. This could be an indirect consequence of growth inhibition, but because of the partial uncoupling between growth and movement in our experiments, this seems unlikely (Fig 1). Another hypothesis comes from recent updates on the growth-defense tradeoff hypothesis. Indeed, extensive growth is thought to be detrimental to the plant under a

pathogen attack because the plant might be more exposed [52]. We propose that this may also be the case for leaf movements. Transiently suppressing leaf circadian movements may be part of a visual apparency strategy limiting herbivore attack. If this were true, the suppression of this protection by shade would be another example of prioritizing light capture in suboptimal conditions over defense. We found that the suppression of the wounding effect by shade depends on classical signaling elements of shade-induced hyponasty, including phyB, PIFs and YUCs [6,7] (Figs 1C, 4C and S3). Interestingly, in tomato, wounding was reported to inhibit elevated temperature-induced leaf hyponasty in a COI1-dependent manner [53]. In that case, under higher ambient temperatures, JA production and signaling upon wounding lead to increased leaf temperatures, decreased photosynthesis and leaf growth. This indicates that although shade-avoidance and thermomorphogenesis share many signaling components and phenotypic outputs [54,55], they might interact differently with JA/wounding pathways.

Previous studies have shown that shaded plants challenged with herbivores, or more generally with pathogens, prioritize growth over defense [18,56]. As reported by others, we also observed a dampening of the JA-dependent wounding response by low R/FR at the gene expression levels (S4D Fig). Under shade-mimicking conditions, less genes were differentially regulated by wounding and among regulated genes many were regulated to a reduced extent compared to control sun condition (S4D Fig). This is consistent with previous studies comparing low R/FR and MeJA treatments [27,57]. For example, genes like *LOX2* or *VSP1*, which are classically induced by herbivores or wounding [58,59], were induced to a lower extent under shade conditions (S4D Fig) [26].

Our transcriptomic approach identified *PKS* genes and especially *PKS4*, as regulators of leaf elevation acting at the interface of shade and wounding pathways (Figs 3 and S5, S6 and S7). *PKS4* expression depends on PIFs upon shade treatment, whereas it is downregulated upon wounding in a MYC-dependent manner (Figs 4 and 5). The importance of regulating *PKS* expression for normal leaf hyponasty is highlighted by altered hyponastic responses in plants over-expressing *PKS* genes or mutants with altered *PKS* expression (Figs 3–5). Interestingly, the *PKS4* promoter was bound by GFP-tagged MYC2 upon MeJA treatment in a ChIP-seq experiment performed with light-grown seedlings [48]. The binding region corresponds to the same one we identified as a PIF4 binding region ("peak" region containing a PBE-box) (Fig 4B). This observation suggests that *PKS4* is regulated differentially by several transcription factors depending on the environmental conditions. Upon combined shade and wounding treatments, there might be a competition between PIFs and MYCs for *PKS4* promoter binding. Since MYC stability is negatively affected by low R/FR [25] and JA signaling is generally attenuated [4], PIFs may have higher chances to bind the *PKS4* promoter in shade conditions explaining why in the combined shade and wound treatment the regulation of *PKS4* expression is dominated by the shade cue (S5 Fig). Such combined regulation by PIF4 and MYC2 might be quite general based on the large overlap of PIF4 and MYC2 binding sites observed in ChIP studies (S9A Fig) [42,48]. Moreover, we found that such an overlap was very common (close to 70%) amongst genes that are induced by shade, repressed by wounding and where the wound effect was suppressed by shade (S9B Fig). We hypothesize that *PKS4* expression is coordinately regulated by MYCs and PIFs to respond to complex environmental situations and propose a model (Fig 6) explaining the regulation of leaf hyponasty in response to shade and wounding (Fig 6).

*PKS* genes function in different light-regulated processes including phototropism [37], hypocotyl growth and orientation [38], leaf positioning and flattening [33,40]. This work identifies a new role of *PKS* genes by showing their specific involvement in the regulation of leaf hyponasty during combined shade and wound treatments. All these responses are based on asymmetric growth. Phototropic bending is a textbook example of differential growth between

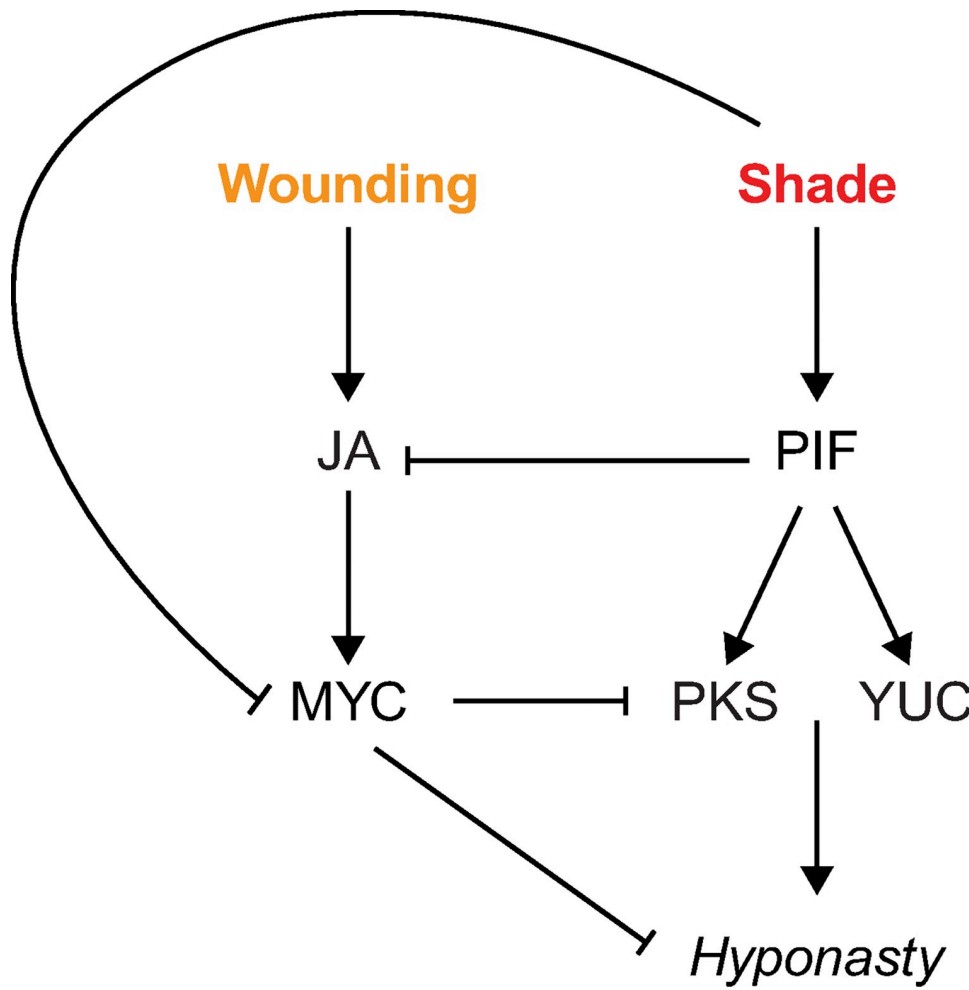

**Fig 6. A model of how shade and wounding may regulate leaf hyponasty.** Shade conditions induce leaf hyponasty through PIF-dependent induction of YUC genes [6,7]. PIFs induce *PKS* expression, but *PKS* are not required for shade-induced hyponasty (under single treatment). In parallel, shade inhibits the JA/wounding pathway through inactivation of JA [26] and inhibition of MYC [24,25]. The JA/wounding pathway inhibits leaf hyponasty and *PKS*. Under combined shade and wounding treatment, shade inhibits the negative effect of wounding on hyponasty through *PKS*.

both sides of a stem/hypocotyl [60,61]. Upward leaf movements have also been proposed to result from differential cell elongation between both sides of the petiole. This hypothesis is notably based on (1) cell length measurements on abaxial vs adaxial sides of hyponastic petioles after ethylene or waterlogging treatment [62,63] and (2) the observation of differential auxin responses between petiole sides in response to low R/FR [7] or elevated temperature [64]. Circadian leaf movements under sun conditions also decrease when leaves age and stop growing [65], therefore strongly linking movement to growth. Interestingly, in seedlings *PKS4* is expressed in the hypocotyl elongation zone [38], where bending occurs upon unidirectional illumination. Here, we show that in rosettes, *PKS4* is also expressed where differential growth is predicted to happen, that is in the petioles of young growing leaves (S5B Fig), which are the ones showing the largest movements at a given developmental stage [32]. Besides, *PKS4* expression is induced by auxin [45], which is known to play a crucial role in leaf hyponasty [6,7] and growth in general. An attractive hypothesis is that shade-induced auxin production

overrides the wound-induced leaf position response. This is consistent with the leaf hyponastic response of *yuc2589* mutants (Fig 1). Therefore, *PKS4* expression pattern correlates with its functions as a regulator of differential growth processes. Although a model explaining hyponasty strictly based on growth is attractive, we note that hyponasty and growth can be partially uncoupled (Fig 1). We previously showed that auxin signaling was particularly important in the vasculature for shade-induced hyponasty [6], where *PKS4* is also expressed (S5B Fig). Moreover, propagation of the wounding signals (both electrical and $Ca^{2+}$-dependent) also appears to rely on both phloem and xylem, especially in variations of pressure inside these tissues [10,16,66]. Variations in leaf hydraulics in response to environmental cues could thus be a tempting candidate mechanism to explain leaf movements. How this may relate to PKS function requires future studies.

## Material and methods

### Plant material

All Arabidopsis lines used in this study are in the Col-0 background, except the original *yuc5* allele which was in La-er [31,67]. The following mutants were described previously: *aos* [30], *myc2myc3myc4* [47], *phyB-9* [68], *pif4pif5pif7* [5], *pks1-1* and *pks2-1* [69], *pks3-9* [40], *pks4-1* and *pks4-2* [37], *yuc2-1yuc5-3yuc8-1yuc9-1* [31,67]. The transgenic lines *CYCB1;1:GUS* (Col-0/ *CYCB1;1pro::NterCycB1;1-GUS*) [70], *JGP* (Col-0/*JAZ10::GUSplus*) [28], PIF4-HA (*pif4-101/ PIF4:: PIF4-3xHA*) [43], *PKS2-OX* (Col-0/*35S::PKS2*) [69], *PKS4-OX* (Col-0/*35S::PKS4*) and *PKS4:GUS* (Col-0/*pPKS4::GUS*) [38] were also already described. Higher order *pks* mutants were obtained by crosses.

### Construction of CRISPR mutant alleles

*pks2-3* mutant allele was obtained by using the CRISPR-Cas9 system as described in [71] but using a modified version of the CRISPR vector in which the hygromycin selection marker was replaced by a seed-specific GFP expression. This plasmid called pEC1.2/EC1.1-SpCas9-GFP is deposited at https://www.addgene.org/161933/. In details, the *pks2-3pks4-2* double mutant was directly obtained by mutating *PKS2* in *pks4-2* plants using two sgRNA directed against *PKS2* genomic sequence: 5'-AGTGATCCAGACTCACCAGA-3' and 5'-GCAAAGCTCGAAGAA TTCCT-3'. The selected mutation corresponds to a 617 bp deletion inside the CDS (starting at base 361 bp after start codon). Genotyping primers sequences are the following: forward primer MT111 5'-CCCAGAAGAACCTAATGAGTGGTATC-3' and reverse primer CF330 5'- AGCTCGGTGTTCTGTTCATG-3'.

### Growth conditions and phenotyping analyses

All experiments were performed under long-day conditions (16h day/8h night). Absolute light intensities were measured with an IL1400A radiometer (https://www.intl-lighttech.com/ product-group/light-measurement-optical-filters) supplemented with a combination of a white diffuser filter and a PAR filter [72]. The R (640–700 nm)/FR (700–760 nm) ratio was measured with an Ocean Optics USB2000+ spectrometer. Temperature was monitored with Thermochron iButtons (Maxim Integrated Products). Seeds were surface sterilized, sown on soil and stratified for 4 days in the dark at 4°C. Plants were grown for 14 days in a culture chamber under 120 µmol.m$^{-2}$.s$^{-1}$ of PAR and 23°C day/18°C night temperature cycles. On day 9, seedlings at the same developmental stage (stage 1.02 [73]) were transferred to individual pots overtopped by a dome of soil. On day 15, plants at stage 1.06 [73] were moved to an E36-L Percival incubator and acclimatized for 1 day to 60 µmol.m$^{-2}$.s$^{-1}$ PAR (R/FR ratio = 1.4)

and constant 20˚C. Leaves were numbered and wounded from the oldest to the youngest. Wounding treatments on leaves 1, 2, and 3 were performed from day 16 to 18 at ZT3, one leaf per day, by crushing half of the blade with toothed metal forceps. On day 18 at ZT3, the lower shelf was supplemented with FR light (14 μmol.m$^2$.s$^{-1}$) to trigger a low R/FR treatment (R/FR ratio = 0.2). Silhouette images of leaf 4 were taken on day 18 at ZT11 with a Canon EOS 550D camera and elevation angles were measured as previously described [6]. For all genotypes, we represented elevation angle as tip elevation angle [6,32], except for the quadruple *pks1234* mutant, for which we measured only the petiole angle because the extreme leaf flattening defect [40] made it difficult to position leaf tip. Petiole length of leaf 4 was measured on day 21 as described [5]. For time-lapse experiments, elevation angle of leaf 4 was assessed as described [6].

## Statistical analyses

For leaf angle and petiole length phenotypes, analysis of variance (ANOVA) were performed using R software using aov function (https://www.r-project.org/). Depending on the experiment, two or three factors were considered (genotype and/or light treatment and/or wounding treatment) and two-way or three-way ANOVA were performed respectively. For each individual replicate experiment, we computed the highest level interaction term and retrieved the corresponding p-value in R. ANOVAs were followed by Tukey's Honest Significance Differences (HSD) test (AGRICOLAE package with default parameters in R).

## GUS staining

*PKS4*::*GUS* rosettes were grown and treated as described above and harvested at ZT9. High R/FR-grown *CYCB1;1*::*GUS* rosettes were harvested at ZT7.5 on days 15 to 17. High R/FR-grown *JGP* rosettes at day 15 were wounded on leaf 1 at different ZT (ZT2, ZT3, ZT4 or ZT5) and all harvested at ZT6. Samples were incubated in 90% acetone overnight at −20˚C. Plants were washed twice in 50 mM sodium phosphate buffer (pH 7.2) and then were vacuum infiltrated for 20 min in 5-bromo-4-chloro-3-indolyl-β-glucuronide (X-gluc) buffer [50 mM NaPO4 (pH 7.2), 0.1% Triton X-100, 0.5 mM K3[Fe(CN)6], and 2 mM X-gluc] and subsequently were incubated at 37˚C for 16 h. Rosettes were cleared overnight in 70% ethanol at 4˚C before being photographed with a Canon EOS 550D camera.

## RT-qPCR

RNA extraction and quantitative RT-PCR were performed as previously described [74]. Data was normalized against two reference genes (*UBC* and *YLS8*) using ΔΔCt method. Gene-specific oligonucleotides used for qPCR reactions are listed in S3 Table.

## ChIP-qPCR

10-d-old PIF4-HA seedlings grown on MS/2 plates in LD, high R/FR, at 21˚C were either kept in high R/FR or shifted at ZT2 to low R/FR for 2h before harvesting in liquid nitrogen. Chromatin extraction and immunoprecipitation were performed as described previously [75]. The qPCR was done in triplicates on input and immunoprecipitated DNA. Oligonucleotides are listed in S3 Table.

## RNA-sequencing and bioinformatics

The quality of the sequencing data was first assessed using *fastqc* (v0.11.2; https://www. bioinformatics.babraham.ac.uk/projects/fastqc/) and *MultiQC* (v1.0) [76]. They were trimmed using *trimGalore* (v0.4.0, q 30—length 40;

https://www.bioinformatics.babraham.ac.uk/projects/trim_galore/) which is a wrapper for *cutadapt* (v1.11) [77] and again verified using *fastqc* and *MultiQC*. The paired reads were mapped onto the TAIR10 genome with *STAR* (v2.5.0b,—outFilterMismatchNmax 4) [78] and reads/gene directly counted using the integrated *STAR* function "GeneCounts" based on the TAIR10 gene annotation.

Differential gene expression analysis was conducted in *R* (v3.4.1) using *limma* (v3.32.5) [79]. Genes which had an average logCPM of < -0.5 CPM were removed to reduce noise and samples normalized using the TMM (trimmed mean of M-values) method from *edgeR* (v3.18.1) [80] prior *limma*'s differential gene expression analysis. Further analysis and comparisons were all conducted in R (https://www.r-project.org/). Heatmaps were generated with *ComplexHeatmap* (v1.14.0, clustering_distance_rows = "pearson") [81]. Venn diagrams were generated using the VENNY 2.1 webtool (https://bioinfogp.cnb.csic.es/tools/venny/).

### Comparisons with published ChIP-seq datasets

Lists of PIF4-bound genes and MYC2-bound genes were obtained from published articles [42,48]. The significance of the overlap between these two lists was determined using a hypergeometric test in R with a significance threshold of $2.5 \times 10^{-4}$.

### Supporting information

**S1 Table. List of 59 candidate genes.**
(XLSX)

**S2 Table. ANOVA three-way interaction term.**
(XLSX)

**S3 Table. Oligos used in the study.**
(XLSX)

**S4 Table. Datapoints for the experiments shown in all figures.**
(XLSX)

**S1 Fig. Description of the experimental system. A)** Time-course GUS activity in rosettes of the *JAZ10*:GUS reporter line upon wounding of leaf 1. 15-day-old rosettes grown in long days under high R/FR conditions were harvested at different times after wounding. Red arrow: wounded leaf. **B)** RT-qPCR analysis of JA-induced genes (*JAZ10*, *LOX2*, *VSP2*) in entire leaf 4 of unwounded and wounded *JAZ10*:GUS plants. 16-day-old rosettes grown in LD, high R/FR, were either touched (control, unwounded) or wounded on leaf 1 at ZT3 and leaves 4 were harvested 3h after treatment. For each gene, expression levels are given relative to housekeeping genes and expression in unwounded plants is arbitrarily set to 1. Error bars correspond to SD from 3 biological replicates. **C)** GUS activity in the *CYCB1;1*:GUS reporter line as a marker for dividing cells. Rosettes grown in long days under high R/FR conditions were harvested from day 15 to 17 after sowing. Leaves are numbered from the oldest to the youngest. Scale bar, 2 mm.
(PDF)

**S2 Fig. Shade and wounding do not interact on petiole elongation.** Petiole lengths of Col-0 with 3 different protocols of shade vs wounding treatments. Top: Schematic of the different experimental procedures. Wounds are performed individually on leaf 1, then leaf 2 and then leaf 3. The shade treatment either starts at the same time as the first wound (left), at the same time as the third wound (middle, see Fig 1A) or one day before the first wound (right). We measured petiole length of leaf 4 at the end of the treatments (day 21). Bottom: Petiole length

phenotype of wt Col-0 plants (n = 12–15 plants/genotype/condition). For each protocol, a representative experiment from 3 biological replicates is presented. Different letters indicate significant differences (Tukey's HSD test following a two-way ANOVA, P < 0.05).
(PDF)

**S3 Fig. Suppression of wounding effect in a *phyB* background.** Leaf angle and petiole length phenotype of wt Col-0 and *phyB-9* mutant plants grown in high R/FR (n = 10 plants/genotype/ condition). As described on Fig 1A, wounds are performed individually on leaf 1 (day 16), then leaf 2, then leaf 3 and plants are then kept in sun condition. We measured elevation angle of leaf 4 at the end of day 18 on pictures of each individual plant and petiole length of the same leaf at the end of the treatments (day 21). Representative experiment from 3 biological replicates. Graphs are represented as violin plots, which present a combination of a box plot and a kernel density plot. In each box plot, the white dot represents the median, black boxes extend from the 25th to the 75th percentile, while the vertical black line extends to 1.5 times the interquartile range of the lower and upper quartiles, respectively. A rotated kernel density plot surrounds each side of the box plot. Different letters indicate significant differences (Tukey's HSD test following a two-way ANOVA, P < 0.05).
(PDF)

**S4 Fig. Wounding-induced gene expression programs. A)** Number of genes significantly differentially regulated by wounding in the different conditions: genotypes (Col-0 and *aos*), timepoints (Time 1 and Time 2) and shade treatment (sun in beige, shade in dark red). Genes are considered differentially expressed when |log2 fold change|>1 and pval < 0.05. **B)** Heatmap with hierarchical clustering of significantly differentially regulated genes by wounding in the different conditions (same genes as in A). **C)** Venn diagram comparing wound-regulated genes (at Time 1 and/or Time 2) in sun vs shade. **D)** Scatterplots of expression (Log2 FC) of wound-regulated genes in sun vs shade at Time 1 (left graph) or Time 2 (right graph). Blue line represents the linear regression of shown data points (equation and regression coefficient are added on the right). Dotted grey line represents y = x.
(PDF)

**S5 Fig. *PKS4* expression is regulated by shade and wounding. A)** *PKS4* expression in RNA-seq dataset (at Time 1, 1h30, wt samples). Values correspond to average log CPM from the three biological replicates. **B)** *PKS4* expression in *PKS4*::GUS reporter line subjected to shade treatment. Rosettes grown in long days under high R/FR conditions were either kept in high R/FR or subjected to low R/FR for 6 hours starting at ZT3 and harvested on day 18 after sowing. Scale bar, 3.6 mm. **C)** Leaf angle and petiole length phenotype of wt Col-0 and *pks4-2* mutant plants (n = 10 plants/genotype/condition). Representative experiment from 2 biological replicates. Experiments were done as described on Fig 1A. Graphs are represented as violin plots, which present a combination of a box plot and a kernel density plot. In each box plot, the white dot represents the median, black boxes extend from the 25th to the 75th percentile, while the vertical black line extends to 1.5 times the interquartile range of the lower and upper quartiles, respectively. A rotated kernel density plot surrounds each side of the box plot. Different letters indicate significant differences (Tukey's HSD test following a three-way ANOVA, P < 0.05).
(PDF)

**S6 Fig. A single *pks2* mutant displays a wild-type elevation response to shade and wounding. A)** *PKS2* expression in RNA-seq dataset (left at Time 1, right at Time 2, wt samples). Values correspond to average log CPM from the three biological replicates. **B)** Leaf angle and petiole length phenotype of wt Col-0 and *pks2-1* mutant plants (n = 10 plants/genotype/

condition). Representative experiment from 2 biological replicates. Experiments were done as described on Fig 1A. Graphs are represented as violin plots, which present a combination of a box plot and a kernel density plot. In each box plot, the white dot represents the median, black boxes extend from the 25th to the 75th percentile, while the vertical black line extends to 1.5 times the interquartile range of the lower and upper quartiles, respectively. A rotated kernel density plot surrounds each side of the box plot. Different letters indicate significant differences (Tukey's HSD test following a three-way ANOVA, $P < 0.05$).
(PDF)

**S7 Fig. The double *pks2-3pks4-2* mutant displays similar responses to shade and wounding than the other allelic combination.** Leaf angle and petiole length phenotype of wt Col-0 and *pks2-3pks4-2* mutant plants (n = 10 plants/genotype/condition). Representative experiment from 2 biological replicates. Experiments were done as described on Fig 1A. Graphs are represented as violin plots, which present a combination of a box plot and a kernel density plot. In each box plot, the white dot represents the median, black boxes extend from the 25th to the 75th percentile, while the vertical black line extends to 1.5 times the interquartile range of the lower and upper quartiles, respectively. A rotated kernel density plot surrounds each side of the box plot. Different letters indicate significant differences (Tukey's HSD test following a three-way ANOVA, $P < 0.05$).
(PDF)

**S8 Fig. Expression of *PKS2* and *PKS4* is repressed by wounding in a JA-dependent manner.** Comparison *PKS2* (**A**, **B**) or *PKS4* (**C**, **D**) expression in *wt* and *aos* samples under sun conditions at Time 1 (**A**, **C**) or Time 2 (**B**, **D**). Values correspond to average log CPM from the three biological replicates.
(PDF)

**S9 Fig. Overlap between PIF and MYC target genes. A)** Venn diagram comparing PIF4-bound genes [42] to MYC2-bound genes [48] (hypergeometric test: $p < 2.5 \times 10^{-4}$). **B)** Venn diagram comparing genes bound by PIF4 and MYC2 (5114 genes from overlap in A) to the 46 genes from the list of candidates (S1 Table) which are up-regulated by shade and down-regulated by wounding (hypergeometric test: $p < 2.5 \times 10^{-4}$).
(PDF)

## Acknowledgments

We thank Mieke de Wit for her great help on the starting of this project. Edward Elliston Farmer (University of Lausanne) and people from his lab for fruitful discussions, technical advice to perform wounding experiments, input on the project as well as for providing *aos* and *JGP* seeds. We thank Philippe Reymond (University of Lausanne) for providing *myc234* seeds. We thank members of the Lausanne Genomic Technologies Facility (GTF) from the University of Lausanne (Switzerland) for performing RNA-sequencing.

## Author Contributions

**Conceptualization:** Anne-Sophie Fiorucci, Christian Fankhauser.

**Data curation:** Emanuel Schmid-Siegert.

**Formal analysis:** Anne-Sophie Fiorucci, Olivier Michaud, Emanuel Schmid-Siegert.

**Funding acquisition:** Christian Fankhauser.

**Investigation:** Anne-Sophie Fiorucci, Olivier Michaud, Emanuel Schmid-Siegert, Martine Trevisan, Laure Allenbach Petrolati, Yetkin Çaka Ince.

**Methodology:** Anne-Sophie Fiorucci, Emanuel Schmid-Siegert, Christian Fankhauser.

**Resources:** Christian Fankhauser.

**Supervision:** Christian Fankhauser.

**Validation:** Anne-Sophie Fiorucci.

**Visualization:** Olivier Michaud, Emanuel Schmid-Siegert.

**Writing – original draft:** Anne-Sophie Fiorucci, Christian Fankhauser.

**Writing – review & editing:** Anne-Sophie Fiorucci, Christian Fankhauser.

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
