## [Decision Letter · Decision Letter 0]

27 Dec 2021

Dear Dr Fankhauser,

Thank you very much for submitting your Research Article entitled 'Shade suppresses wound-induced leaf repositioning through a mechanism involving PHYTOCHROME KINASE SUBSTRATE (PKS) genes.' to PLOS Genetics.

The manuscript was fully evaluated at the editorial level and by independent peer reviewers. The reviewers recognized that this study constitutes a significant advance. They, nevertheless, raised some substantial concerns about the current manuscript. While the involvement of  PSKs in the interplay between wounding and shade avoidance was clearly demonstrated, this study would benefit from additional mechanistic insights into how these genes ultimately influence the hyponastic response. Some of the reviewers also thought that the conclusions concerning the regulation of PSKs by MYB were not fully supported. These conclusions could perhaps be recast as hypotheses to ensure that the readers do not misinterpret them. Additional details on the statistical analyses should also be provided. The impact of the study would be better demonstrated by proposing a model integrating the results into the current knowledge within the field. Based on the reviews, we will not be able to accept this version of the manuscript, but we would be willing to review a much-revised version. We cannot, of course, promise publication at that time.

If you decide to revise the manuscript for further consideration at PLOS Genetics, please aim to resubmit within the next 60 days, unless it will take extra time to address the concerns of the reviewers, in which case we would appreciate an expected resubmission date by email to plosgenetics@plos.org.

[LINK]

We are sorry that we cannot be more positive about your manuscript at this stage. Please do not hesitate to contact us if you have any concerns or questions.

Yours sincerely,

Adrien Sicard

Guest Editor

PLOS Genetics

Claudia Köhler

Section Editor: Plant Genetics

PLOS Genetics

Reviewer's Responses to Questions

**Comments to the Authors:**

Reviewer #1: Plants alter their development and growth in response to both abiotic and biotic stress. The relationship between biotic and abiotic responses and the mechanisms integrating both types of stresses simultaneously are still poorly understood. For example, shade and wounding by herbivores elicit opposing effects on tissue/organ growth. Plants respond to shade or the presence of competing neighbors by promoting petiole and stem elongation as well as triggering an upward leaf movement (hyponasty), thereby escaping shade. In contrast, during herbivore attack, the plant hormone Jasmonate (JA) elicits wounding responses that inhibit tissue growth and leaf movement. While the impact of shade on JA responses has been well studied, the mechanism that integrates the two types of signals remains elusive.

Here the authors explored the relationship between shade and wounding responses in terms of leaf/petiole growth and leaf hyponasty. The authors carefully devised a protocol to precisely quantify the effects of wounding and shade on a specific intact leaf. The authors show that, when challenged by both shade and wounding, Arabidopsis is able to sustain the shade-mediated leaf hyponasty but not leaf elongation, indicating that leaf growth and movement can be uncoupled in the combined shade and wounding treatments. Global transcriptomic analysis revealed that the shade treatment reduced the impact of wounding on gene expression. The RNA-seq analysis also revealed a group of genes whose expression patterns match the leaf movement response in the shade, wounding, and combined shade/wounding treatments; these include PSK4 and, to a less degree, PSK2. The authors then present genetic evidence supporting that PKS4 and PKS2 act redundantly in the shade-dependent suppression of the wound-induced inhibition of leaf hyponasty. Interestingly, both PKS4 and PKS3 are induced directly by PIF4/5/7 in response to shade. PIF4/5/7, similar to YUCs, are required for the lack of the wounding effect on hyponasty in shade. Moreover, PKS4 and PKS2 are repressed by wounding via MYCs; a myc234 triple mutant lost JA-induced inhibition of leaf hyponasty.

Overall, this elegantly-designed study provides compelling genetic evidence showing distinct relationships between shade and wounding on leaf/petiole growth and leaf hyponasty. More importantly, the results reveal novel insight into the genetic elements required for the shade-dependent repression of the wound-induced inhibition of leaf hyponasty -- namely PIFs, auxin, PKS4/2 -- and demonstrate the function of PKS4/2 in both the positive and negative regulation of leaf hyponasty by shade and wounding, respectively. The manuscript is well written, and the figures are beautifully presented.

My only suggestion is that the authors might want to elaborate the part in Discussion describing the hypothesis about the competition of PIFs and MYCs in PKS4/2 expression. The results in Figure S3 demonstrate that the wound-induced inhibition of leaf hyponasty depends on phytochrome B (phyB). Studies from the authors’ lab and others have shown that phyB inhibits the stability and activity of PIFs. Together, the current data support the model that, in normal light conditions (without shade), phyB inhibits PIFs to allow MYCs function prominently in wounding to repress PKS4/2 expression, whereas, in shade condition, the attenuation of phyB functions enhances the amount and activity of PIFs, which can antagonize MYCs in wounding to promote PKS4/2 expression.

Reviewer #2: Shade suppresses wound-induced leaf repositioning through a mechanism involving PHYTOCHROME KINASE SUBSTRATE (PKS) genes

Fiorucci et al. studied the interaction of wounding and shade avoidance in Arabidopsis to understand how the light signalling and wound-induced JA-signalling are integrated. First, consistent with earlier studies, they find that when individually applied shade induces a hyponasty response while mechanical wounding leads to lowering of leaf position. When applied together, wound-induced lowered leaf position response was suppressed by shade. While wound-induced response required JA, shade-induced responses required auxin as shown by mutant analyses. They further studied gene expression to understand the molecular basis of the interaction and identified a set of genes showing contrasting regulation by wound and shade and that the wound effect is suppressed by shading. The authors identified PHYTOCHROME KINASE SUBSTRATE 4 (PKS4) for further studies. Using loss of function mutants and transgenic overexpression experiments, the authors find that PKS4 and close homologs are involved in the control of leaf hyponasty in response to wound and shade. They further show that PKS4 and PKS2 are PIF target genes by studying gene expression and chromatin binding of PIF4 at PKS4. The authors also argue that PKS2 and 4 are repressed by MYC transcription factors.

The experiments are well planned and brilliantly carried out with a sound statistical analysis. In general, the conclusions drawn are justified except a few exceptions. For example, the last section of MYC repression of PKS2 and 4 are not convincing. The provided data is not sufficient to support this argument.

While the data shown here in general are well-supported, the major question remains unanswered - how do PKS proteins control the hyponasty response and how are the contrasting signals integrated? To this reviewer, more in-depth analysis on the mechanism of shade-induced suppression of wound-induced leaf positioning through PKS would have been helpful.

What is the biological significance of the transient inhibition of leaf elevation by wounding? The authors argue that effect on leaf positioning by wounding is more than just a consequence of a general growth suppression and perhaps be of an adaptive value. This needs to be further supported. It could have helped if the role of PKS in plant defence under shade conditions were investigated.

To this reviewer, while this is an interesting and well executed study, the findings are rather incremental and stops short of making a significant advance.

Reviewer #3: Review of Fiorucci et al for PLoSG

This manuscript explore the relationship between shade and defense signaling in the control of plant morphology. Although the effects of shade and defense signaling interactions on defense responses have been explored in a number of prior publications, the effect of these interactions on growth and morphology is much less explored. Thus, this manuscript is novel and is a significant contribution to the field. This manuscript describes the molecular-genetic mechanisms underlying the effect of shade/defense interactions on leaf angle, and reveals that genes in the PKS family are an important intersection point for these two pathways. The manuscript is well written and conclusions are well supported. There are some issues that need to be addressed before considering publication.

Major:

1) The authors report p-values for the interaction term in two-way ANOVA tests but not in three-way ANOVA tests (e.g. genotype:light:wounding for fig 1BC, 3AB, etc). In theory the three-way interaction term from an ANOVA or linear regression would be the best test for whether or not a mutant affects the light:wounding interaction, but I acknowledge that it can be difficult to achieve enough power to adequately test this term in practice, potentially leading to false negatives. Another approach might be to fit separate two-way linear regressions for WT and mutant, obtain 95% confidence intervals on the interaction term and determine if they overlap or not. Current conclusions are based on (I think) Tukey’s HSD but this is not completely clear. If the authors stick with the Tukey HSD approach it would be helpful if they explained how they are interpreting these tests to arrive at their conclusions (along the lines of “shade caused a similar increase in leaf angle in unwounded Col-0 and pks2pks4 (Tukey HSD group a) but this effect was reduced in the pks2pks4 double mutant (Tukey HSD group b for Col-0 vs group c for pks2pks4)”. They is probably a more elegant way to write this, but hopefully that gets the idea across. The reason that the regression or ANOVA approach is favored over the Tukey HSD is that the term that we want to test is related to differences in the wounding response, which is only indirectly addressed by the Tukey approach. In the end I am OK with the Tukey approach if it is better explained, but I suggest that the authors at least consider the other approaches I have outlined if they have not already. On a related note, the figure legends and methods state that a three-way ANOVA was performed. The methods should describe the formula for that (were all two and three way interaction terms used?)

2) The authors conclude that the transcriptional response to wounding is more rapid than the response to shade (line 218) but I do not think that the data supports this, because the first wounding treatment is 2 days earlier than the shade treatment (Fig 1A).

Minor:

Lines 152-183. This is a long paragraph that is a bit difficult to follow, especially because the authors go back and forth between the different responses that they are measuring are not always clear. For example, lines 167-168 “We concluded that there is no interaction shade and wounding…” this is for petioles but that is not indicated in this sentence or the one above. And in this section of the paragraph petiole length is discussed before leaf angle even though they are in the opposite order in the figure. Anyway: the authors should consider whether this section can be organized in a more clear manner; it is easy for the reader to get lost.

Fig 2B: the order of the timepoints in the heat map is reversed that from 2A (and from the progression of time) and should be flipped. In the worst case the heat map could be flipped and relabeled “by hand” (outside of R) if it is not possible to have R present time 1 on the left.

Fig S4B: are the genotype labels wrong? If not, why does aos have a stronger response than Col on the heatmap?

Fig 3B y-axis label (left plot) is “Petiole angle” but elsewhere “Leaf angle” is used.

Fig 4B are “peak” and “control” mislabeled in the bar plot? Also PKS4 doesn’t have solid evidence for induction by shade. Could say marginal in text.

qPCR methods: although a citation to a previous paper is given, it would be nice if a sentence or two could be added about the analysis (is this delta-delta CT?) so that the curious does not have to flip back to the previous paper.

**Have all data underlying the figures and results presented in the manuscript been provided?**

Reviewer #1: Yes

Reviewer #2: Yes

Reviewer #3: **No: **Numerical data underlying graphs is not provided.

PLOS authors have the option to publish the peer review history of their article (what does this mean?). If published, this will include your full peer review and any attached files.

Reviewer #1: **Yes: **Meng Chen

Reviewer #2: No

Reviewer #3: No

---

## [Decision Letter · Decision Letter 1]

20 Apr 2022

Dear Dr Fankhauser,

We are pleased to inform you that your manuscript entitled "Shade suppresses wound-induced leaf repositioning through a mechanism involving PHYTOCHROME KINASE SUBSTRATE (PKS) genes." has been editorially accepted for publication in PLOS Genetics. Congratulations!

Yours sincerely,

Adrien Sicard

Guest Editor

PLOS Genetics

Claudia Köhler

Section Editor: Plant Genetics

PLOS Genetics

Comments from the reviewers (if applicable):

Reviewer's Responses to Questions

**Comments to the Authors:**

Reviewer #2: The revised manuscript and the accompanying response to reviewers have satisfactorily addressed most of the concerns raised by this reviewer. The study indeed identifies an interesting phenomenon. The molecular mechanism and biological significance of the reported phenomenon will be interesting questions to be addressed.

Reviewer #3: My comments have been adequately addressed.

**Have all data underlying the figures and results presented in the manuscript been provided?**

Reviewer #2: Yes

Reviewer #3: **No: **Raw data of plant measurements is not available (or I cannot find it)

PLOS authors have the option to publish the peer review history of their article (what does this mean?). If published, this will include your full peer review and any attached files.

Reviewer #2: No

Reviewer #3: **Yes: **Julin N Maloof

**Data Deposition**

http://datadryad.org/submit?journalID=pgenetics&manu=PGENETICS-D-21-01498R1

**Press Queries**

---

## [Editor Report · Acceptance letter]

23 May 2022

PGENETICS-D-21-01498R1 

Shade suppresses wound-induced leaf repositioning through a mechanism involving *PHYTOCHROME KINASE SUBSTRATE (PKS)* genes. 

Dear Dr Fankhauser, 

We are pleased to inform you that your manuscript entitled "Shade suppresses wound-induced leaf repositioning through a mechanism involving *PHYTOCHROME KINASE SUBSTRATE (PKS)* genes." has been formally accepted for publication in PLOS Genetics! Your manuscript is now with our production department and you will be notified of the publication date in due course.

With kind regards,

Anita Estes

PLOS Genetics

On behalf of:
